# Effects of Angiotensin 1-7 and Mas Receptor Agonist on Renal System in a Rat Model of Heart Failure

**DOI:** 10.3390/ijms241411470

**Published:** 2023-07-14

**Authors:** Ravit Cohen-Segev, Omri Nativ, Safa Kinaneh, Doron Aronson, Aviva Kabala, Shadi Hamoud, Tony Karram, Zaid Abassi

**Affiliations:** 1Department of Physiology and Biophysics, Rappaport Faculty of Medicine, Technion-Israel Institute of Technology, Haifa 31096, Israel; msc12@gmail.com (R.C.-S.); safakinaneh@gmail.com (S.K.); avivak@technion.ac.il (A.K.); 2Department of Urology, Rambam Health Center, Haifa 3109601, Israel; o_nativ@rambam.health.gov.il; 3Cardiology, Rambam Health Care Campus, Haifa 3109601, Israel; d_aronson@rambam.health.gov.il; 4Department of Internal Medicine E, Rambam Health Care Campus, Haifa 3109601, Israel; s_hamoud@rambam.health.gov.il; 5Vascular Surgery, Rambam Health Care Campus, Haifa 3109601, Israel; t_karram@rambam.health.gov.il; 6Laboratory Medicine, Rambam Health Care Campus, Haifa 31096, Israel

**Keywords:** heart failure, kidney, ACE2, angiotensin 1-7, MasR

## Abstract

Congestive heart failure (CHF) is often associated with impaired kidney function. Over- activation of the renin–angiotensin–aldosterone system (RAAS) contributes to avid salt/water retention and cardiac hypertrophy in CHF. While the deleterious effects of angiotensin II (Ang II) in CHF are well established, the biological actions of angiotensin 1-7 (Ang 1-7) are not fully characterized. In this study, we assessed the acute effects of Ang 1-7 (0.3, 3, 30 and 300 ng/kg/min, IV) on urinary flow (UF), urinary Na^+^ excretion (UNaV), glomerular filtration rate (GFR) and renal plasma flow )RPF) in rats with CHF induced by the placement of aortocaval fistula. Additionally, the chronic effects of Ang 1-7 (24 µg/kg/h, via intra-peritoneally implanted osmotic minipumps) on kidney function, cardiac hypertrophy and neurohormonal status were studied. Acute infusion of either Ang 1-7 or its agonist, AVE 0991, into sham controls, but not CHF rats, increased UF, UNaV, GFR, RPF and urinary cGMP. In the chronic protocols, untreated CHF rats displayed lower cumulative UF and UNaV than their sham controls. Chronic administration of Ang 1-7 and AVE 0991 exerted significant diuretic, natriuretic and kaliuretic effects in CHF rats, but not in sham controls. Serum creatinine and aldosterone levels were significantly higher in vehicle-treated CHF rats as compared with controls. Treatment with Ang 1-7 and AVE 0991 reduced these parameters to comparable levels observed in sham controls. Notably, chronic administration of Ang 1-7 to CHF rats reduced cardiac hypertrophy. In conclusion, Ang 1-7 exerts beneficial renal and cardiac effects in rats with CHF. Thus, we postulate that ACE2/Ang 1-7 axis represents a compensatory response to over-activity of ACE/AngII/AT1R system characterizing CHF and suggest that Ang 1-7 may be a potential therapeutic agent in this disease state.

## 1. Introduction

Despite the advances in understanding the pathophysiology and application of appropriate treatments in recent decades, congestive heart failure (CHF) still imposes a grim prognosis as about half of the patients are expected to die within 5 years of diagnosis [1]. The pathogenesis of CHF involves complex cardiac, renal and neurohormonal alterations. At the cardiovascular level, the main manifestation is cardiac hypertrophy, which involves cellular, histological and functional changes, aimed at allowing the heart to compensate functionally to overload or injury [2,3]. Although cardiac remodeling is initially a beneficial and adaptive process, progressive functional changes eventually result in maladaptive and irreversible alterations.

The kidney plays a major role in the pathophysiology of CHF [4,5]. Impaired renal function and reduced glomerular filtration rate (GFR) are considered to be a strong independent predictor of mortality in patients with CHF [6,7]. Several neurohumoral systems with renal vasoconstrictors are thought to mediate the alterations in renal function in CHF [4,5]. Specifically, the sympathetic nervous system (SNS), vasopressin, endothelin and the renin–angiotensin–aldosterone system (RAAS) are activated in CHF [8,9,10,11,12]. Concomitant with the stimulation of the vasoconstrictor neurohumoral systems, compensatory vasodilatory/natriuretic systems are also activated in CHF, serving to counterbalance the actions of the opposing vasoconstrictor systems. Among these vasodilatory/natriuretic agents, those particularly studied in both patients and animals with heart failure are the natriuretic peptides, primarily atrial natriuretic peptide (ANP), brain natriuretic peptide (BNP) and the nitric oxide (NO) system [13,14]. Numerous studies in patients and in experimental models of CHF have established the important role of the RAAS in the progression of cardiovascular and renal dysfunction in CHF [15,16]. Prolonged activation of the RAAS has direct deleterious actions on the myocardium, independent of its systemic hemodynamic effects [17,18]. In this regard, both angiotensin II (Ang II) and aldosterone are known for their stimulatory effects on myocyte hypertrophy, fibrosis and apoptosis, leading ultimately to progressive remodeling and further deterioration in cardiac performance [19]. In light of the deleterious renal and cardiac actions of the classic RAAS, it is not surprising that angiotensin converting enzyme (ACE) inhibitors, Ang II receptor antagonists and aldosterone blockers are cardinal treatment of CHF and progressive kidney diseases [15,19,20,21]. Research in the past three decades has revealed that beside the circulating RAAS, various tissues also express a complete set of renin and angiotensin metabolizing components (tissue RAAS) involved in paracrine cardiovascular and renal regulation [9,22,23]. Moreover, these components were recently identified within cells, thus constituting an intracrine RAAS for regulation of various cellular functions [22,23]. In addition, recent studies have further elaborated the functions of differentially metabolized angiotensin-derived peptides and discovered even new RAAS components [22,23]. One of these relatively novel systems of emerging significance is the angiotensin converting enzyme 2 (ACE2)–Angiotensin 1-7 (Ang 1-7)–Mas system. ACE2 was identified as a homologue of ACE with carboxypeptidase activity that converts the octapeptide Ang II to the heptapeptide Ang 1-7, or alternatively converts angiotensin I to angiotensin 1-9, which becomes Ang 1-7 upon C-terminal dipeptide cleavage by ACE. ACE2 expression is present in various tissues; however, it is maximal in vascular endothelium, kidney, intestine, heart, lung and testis [23]. The biologically active Ang 1-7 exerts its effects through the G-protein-coupled Mas receptor. ACE2–Ang 1-7 –MasR system is an endogenous counter-regulatory system within the RAAS, as Mas receptor agonists or exogenous Ang 1-7 exert effects that are opposite to Ang II. These include vasodilatation, improvement of endothelial function, anti-hypertrophic, anti-fibrotic and antithrombotic effects [23,24]. Ang 1-7 exerts diuretic and natriuretic effects upon infusion into rat kidneys through a direct tubular effect that is independent of vascular tone changes and may involve the proximal tubule [25]. In addition, direct ACE2 activation improves endothelial dysfunction in diabetic and hypertensive rat vessels [24]. Furthermore, a non-peptide MasR agonist was found to improve cardiac remodeling and function in rat models of post-MI CHF and chronic beta-adrenoreceptor stimulation-induced cardiac remodeling [26,27].

The purpose of this study was to investigate the acute and chronic effects of Ang 1-7 or AVE 0991, MasR agonist [28], on renal and cardiac function in rats with CHF, using an aortocaval fistula (ACF) model, which is a well-established and thoroughly characterized model of volume overload in rats [29].

## 2. Results

### 2.1. Acute Protocols

#### 2.1.1. Effects of Ang 1-7 and AVE 0991 on Urine Flow and Na Excretion

Administration of low doses of Ang 1-7 and AV E0991 into control group resulted in ~2–2.5 fold elevation in urinary flow (*p* < 0.01 and *p* < 0.05, respectively) (Figure 1A,B). This diuretic effect was less prominent following infusion of high doses of these compounds. In contrast, administration of incremental doses of either Ang 1-7 or AVE 0991 into CHF rats did not yield significant changes in urine flow (Figure 1A,B). In general, administration of Ang 1-7 or AV E0991 into sham controls resulted in natriuretic response as was evident by elevation of urinary Na^+^ excretion by 2–3 fold compared to baseline values (Figure 1C,D). The natriuretic effect of Ang 1-7 was detected following infusion of all applied doses. Similarly, although to a lower extent, AVE 0991 infusion induced natriuretic response, which reached ~2 fold of basal UNaV in all applied doses (Figure 1D). As expected, basal UNaV was lower in CHF rats as compared with their sham controls. In contrast to sham-operated animals, CHF rats displayed a blunted natriuretic response to Ang 1-7 (Figure 1C). Furthermore, CHF animals exhibited an attenuated natriuretic response to the administration of AVE 0991 in all applied doses (Figure 1D). When FENa% change was calculated, a similar pattern of natriuretic alterations was obtained in both sham- and CHF-treated animals (Figure 1E,F).

#### 2.1.2. Effects of Ang 1-7 and AVE 0991 on Kidney Function

The effects of Ang 1-7 and AVE 0991 on GFR are depicted in Figure 2. Administration of Ang 1-7 or AVE 0991 into sham group resulted in elevation of GFR by 37.5% and 47.7%, respectively (Figure 2A,B). This effect was seen mainly with low dose of the compounds. Infusion of Ang 1-7 into sham controls induced similar changes in RPF as in GFR, where RPF slightly increased in response to all used doses (Figure 2C). Basal GFR and RPF were significantly lower in CHF rats (Figure 2A,C). In contrast to sham controls, CHF animals displayed a blunted response, where both Ang 1-7 and AVE 0991 did not cause any changes neither in GFR nor RPF following infusion of the applied doses (Figure 2A–D).

#### 2.1.3. Effects of Ang 1-7 and AVE0991 on MAP

The impact of Ang 1-7 and AVE 0991 on MAP in both CHF and control rats is presented in Figure 2E,F. Intravenous infusion of either Ang 1-7 or AVE 0991 cause a slight decrease in blood pressure in sham controls. These changes were approximately similar across all doses of Ang 1-7 and AVE 0991 that were applied. Basal MAP in CHF rats was significantly lower than that measured in their sham controls, yet administration of either Ang 1-7 or AVE 0991 did not results in hypotensive effect even at high doses (Figure 2E,F).

#### 2.1.4. Effects of Ang 1-7 and AVE 0991 on Urinary cGMP

Normalized urinary cGMP to GFR values of both CHF and their control rats are presented in Figure 3. Administration of Ang 1-7 to sham-operated animals resulted in significant elevation in urinary cGMP excretion throughout the experiment, especially at lower doses of Ang 1-7 (Figure 3A). However, administration of AVE 0779 into sham controls did not induce significant increase in urinary cGMP/GFR (Figure 3B). It should be emphasized that baseline UcGMP in CHF rats was significantly higher as compared with their sham controls (~5-fold increase). Infusion of either Ang 1-7 or AVE 0779 into CHF subgroup did not further increase UcGMP beyond the elevated baseline levels, except when high dose of Ang 1-7 was applied (Figure 3A,B).

### 2.2. Chronic Protocols

#### 2.2.1. Kidney Function and Kidney Weight

##### Effects of Ang 1-7 and AVE 0991 on UF, U_Na_V and U_K_V in CHF Rats and Sham Controls

Figure 4 depicts the chronic effects (28 day) of Ang 1-7 and AVE 0991 on cumulative UF, UNaV and UKV in sham and CHF rats. During baseline period (5 days prior to CHF induction and initiation of treatment), all groups showed similar UF, UNaV and UKV. Continuous infusion of Ang 1-7 or its mimetic into sham controls did not result in significant change in daily UF or UNaV as compared with those obtained following administration of vehicle alone (Figure 4A,C,D). In contrast, cumulative UF of CHF rats chronically treated with Ang 1-7 was increased as compared with CHF animals treated with vehicle (Figure 4B). Similarly, AVE 0991 increased cumulative UF in CHF rats (Figure 4B). Rats with CHF displayed reduced UNaV as compared with sham controls (Figure 4D). Ang 1-7 treated CHF rats exhibited increased UNaV throughout the treatment period. The slope representing UNaV curve of CHF rats that were chronically treated with Ang 1-7 was significantly higher than that of CHF animals treated with vehicle (*p* < 0.05, ANOVA 2). The natriuretic effect of Ang 1-7 was evident on day 25 (*p* < 0.05) and lasted until the end of the experiment (*p* < 0.01). CHF rats treated with AVE 0991 displayed similar trend (Figure 4D). Specifically, cumulative UNaV values of AVE 0079-treated CHF rats were significantly higher than those obtained in CHF rats treated with vehicle alone where it reached to comparable levels of UNaV measured in sham controls. The increases in UNaV were significant from day 26 until the termination of the experiment (*p* < 0.05) (Figure 4D).

In addition to the natriuretic response, cumulative UKV values increased in Ang 1-7 or AVE 0079-treated CHF as compared with CHF animals treated with vehicle alone (Figure 4,F). Specifically, rats chronically treated with Ang 1-7 significantly increased UKV on the 24th day (*p* < 0.05) and from the 25th day until the end of the experiment (*p* < 0.01). Similar results were observed with AVE 0991. In this regard, rats chronically treated with AVE 0991 significantly increased their cumulative UKV values on the 25th day (*p* < 0.05) and later (Figure 4F). In contrast to CHF rats, chronic administration of either Ang 1-7 or AVE 0079 in sham-operated animals did not affect UKV (Figure 4E).

##### Effect of Ang 1-7 and AVE 0991 on Kidney Weight and Serum Creatinine (sCr)

Kidneys from the various experimental groups were harvested on the last day, weighted and kidney weight to body weight ratio was calculated (Figure 5A,B). The CHF kidneys were slightly lighter than those of control kidneys (1.1 ± 0.05 vs. 1.2 ± 0.04 gr, *p* = 0.13). Overall, no substantial change in absolute kidney weight or KW/BW% was observed in both sham and CHF subgroups following treatment with either Ang 1-7 or AVE 0991 (Figure 5A,B).

sCr levels were determined in all groups chronically treated with Ang 1-7, AVE 0991or vehicle two weeks and four weeks after initiation of treatment (Figure 5C). Basal sCr levels of the vehicle-treated sham controls and CHF rats (2W) were 0.76 ± 0.12 and 2.01 ± 0.27 mg% (*p* < 0.001), respectively. Interestingly, CHF rats chronically treated with either Ang 1-7 or AVE 0991 for 2 weeks exhibited lower sCr than those observed in the vehicle group (*p* < 0.001) (Figure 5C). sCr in all CHF groups were comparable following 4 weeks of treatment. Concerning sham control group, no significant changes were observed after two weeks of treatment. In contrast, after 4 weeks of treatment, sham controls receiving either Ang 1-7 or its mimetic exhibited lower sCr, yet not significant as compared with vehicle-treated group (Figure 5C).

Urine samples collected on the last day of the experiment (4 weeks after the beginning of treatment) were tested for UcGMP levels (Figure 5D). All sham-operated rats exhibited comparable values of UcGMP*UF, regardless of presence or absence of treatment. CHF rats treated either with Ang 1-7 or its agonist exhibited higher levels of UcGMP*UF (2–3 fold) than CHF rats treated with vehicle (Figure 5D). In contrast to the higher basal UcGMP obtained in CHF as compared with sham controls of acute protocols, basal UcGMP in chronic protocol was comparable in vehicle-treated CHF rats and their sham controls.

#### 2.2.2. Cardiac Parameters

##### Effect of Ang 1-7 and AVE 0991 on Cardiac Remodeling

Hearts from the various experimental groups were harvested on the last day of the experiment, weighted and heart weight to body weight ratio (HW/BW%) was calculated (Figure 6A,B). Heart weights of the CHF rats were significantly higher than those of the sham controls (1.6 ± 0.11 vs. 1.19 ± 0.003 g, *p* < 0.05; 1.50 ± 0.16 vs. 1.13 ± 0.03 g, *p* < 0.05; 1.69 ± 0.17 vs. 1.05 ± 0.013 g, *p* < 0.001) (Figure 6A). Ang 1-7-treated CHF rats have a lower HW/BW% ratio as compared with vehicle-treated CHF rats (0.34 ± 0.02% vs. 0.41 ± 0.03%, *p* < 0.05). In contrast, AVE 0991 treatment did not affect HW/BW% of CHF rats (Figure 6B).

##### Effect of Ang 1-7 and AVE 0991 on Plasma BNP

Plasma BNP levels were measured two and four weeks after commencement of treatment (Figure 6C). After two weeks, baseline plasma BNP levels in untreated CHF and control rats were comparable. Both CHF and control rats that were chronically treated with Ang 1-7 have higher BNP levels as compared with their vehicle-treated control groups (Figure 6C). The average value of plasma BNP levels in Ang 1-7-treated sham control rats was 1.7 fold the plasma BNP value of CHF rats that underwent the same treatment (Figure 6C). While administration of AVE 0991 into sham controls for 2 weeks did not affect plasma BNP concentrations, CHF rats exhibited an increase of 2.4 fold following AVE 0991 treatment. At four weeks, the vehicle-treated CHF rats have 4.64 times higher average plasma BNP levels than the vehicle-treated sham controls (Figure 6C). While chronic administration of Ang 1-7 did not affect plasma BNP levels in either sham controls or CHF animals, AVE 0991 increased plasma BNP levels in the CHF subgroup (Figure 6C). Specifically, Ang 1-7-treated CHF rats for 4 weeks have 3.5 times higher BNP than Ang 1-7-treated control animals (*p* < 0.01). The CHF rats that received AVE 0991 have 2.4 times higher BNP values than their vehicle-treated CHF animals (*p* < 0.001) and 5.78 fold those obtained in AVE 0991-treated sham controls (Figure 6C).

#### 2.2.3. Effect of Ang 1-7 and AVE 0991 on RAAS Status

The average aldosterone levels in the vehicle-treated CHF group was 2.4 times that of the vehicle-treated sham controls after two weeks, and 4.5 times after four weeks (*p* < 0.01 for 2 weeks and *p* < 0.001 for 4 weeks) (Figure 7A). Treatment with Ang 1-7 and AVE 0991 for two weeks resulted in substantial decrease in plasma levels of aldosterone in CHF rats (Figure 7A), where the obtained levels were comparable to those obtained in sham controls. As mentioned above, untreated CHF animals displayed even further increase in plasma levels of aldosterone after four weeks from the induction of CHF as compared with CHF animals at 2 weeks. Treatment for 4 weeks with AVE 0991, but not with Ang 1-7, reduced the elevated aldosterone levels in CHF rats. In contrast, following four weeks of Ang 1-7 treatment, the plasma aldosterone levels of the sham control groups were slightly higher than the vehicle-treated sham control animals (Figure 7A).

Plasma Ang II levels were measured in all sham controls and CHF animals two weeks and four weeks after the initiation of treatment (Figure 7B). No significant changes were observed in plasma Ang II following two weeks of the various treatments in sham control rats. Interestingly, circulatory Ang II levels in CHF rats were not elevated as compared with sham controls following two weeks. Furthermore, CHF rats treated with vehicle, Ang 1-7 or AVE 0991 exhibited similar Ang II levels to those observed with their comparable controls.

## 3. Discussion

In agreement with our previous findings, rats with A-V fistula, an experimental model of CHF, displayed impaired kidney function, cardiac hypertrophy and activation of compensatory vasoconstrictor/anti-natriuretic systems [29]. Specifically, untreated CHF rats displayed lower UF, UNaV GFR, RPF and MAP than their sham controls. Acute infusion of either Ang 1-7 or its agonist, AVE 0991, into CHF rats did not significantly affect UF, UNaV, GFR, RPF and UcGMP. In contrast, chronic administration of Ang 1-7 and AVE 0991 exerted significant diuretic, natriuretic and kaliuretic effects in CHF rats, but not in sham controls. SCr and aldosterone levels were significantly higher in vehicle-treated CHF rats as compared with controls. As expected, treatment with Ang 1-7 and AVE 0991 reduced these parameters to comparable levels as observed in sham controls. Furthermore, chronic administration of Ang 1-7 to CHF rats reduced cardiac hypertrophy, a hallmark feature of heart failure. Collectively, chronic administration of Ang 1-7 exerts beneficial renal and cardiac effects in rats with CHF. Thus, we postulate that ACE2/Ang 1-7 axis represents a compensatory response to over-activity of ACE/AngII/AT1R system, suggesting that Ang 1-7 may be a potential therapeutic agent in CHF.

### 3.1. Acute Protocol

This protocol was designed to investigate the acute effects of Ang 1-7 and its agonist AVE 0991 on renal function in experimental CHF. As opposing AT-II, Ang 1-7 has been reported to induce vasodilation, antiproliferation, anti-angiogenesis and anti-hypertrophy, via activation of its MasR [30,31,32]. Acute infusion of either Ang 1-7 or AVE 0991 into sham controls but not CHF rats evoked beneficial renal effects, including increases in UF, UNaV, GFR and RPF (Figure 1, Figure 2 and Figure 3). Specifically, infusion of low-dose Ang 1-7 greatly increased UF, with a gradual reduction of the effect with high doses of the peptide. Similarly, sham controls treated with Ang 1-7 or its agonist AVE 0991 displayed enhancement of the GFR and RPF. Surprisingly, this increase was most prominent at low doses, but not with the high doses of Ang 1-7 or AVE 0079. This could be attributed to the decline in blood pressure and volume depletion, which occurred after increased urine flow in the initial phase of the acute protocol. Urinary flow in sham controls treated with AVE 0991 showed similar trend. These effects correspond with the fact that Ang 1-7, ACE2, and MasR are abundant in the renal tubular epithelium as was revealed by immunostaining analysis [33].

As expected, CHF rats exhibited reduced renal function compared to their sham controls as was evident by lower UNaV, GFR and RPF. The main player of RAAS, Ang II, constricts the afferent and efferent arterioles and increases sensitivity to tubuloglomerular feedback mechanism. These actions elicit decreases in RPF and GFR [34,35]. The differences between the renal responses of CHF rats and sham controls to Ang 1-7 and AVE 0991 could be explained by the fact that CHF rats have lower basal renal perfusion due to activation of vasoconstrictive systems. Ang 1-7 exerts its vasodilatory effect on the afferent arterioles [33], which increases the intraglomerular pressure and thus improves GFR without affecting RPF especially in CHF animals. In contrast, vasodilation of the afferent and to a lesser extent of the efferent arterioles in sham control rats may be responsible for the initial increase in RPF and GFR in these animals. Over time, the capillary glomerular pressure decreases, resulting in reduction of GFR despite normal RPF. These findings are partially in line with those reported by DelliPizzi et al. [36], where infusion of Ang 1-7 into rats increased urine volume mostly at the beginning but also throughout the experiment. These animals have also demonstrated great increments of UNaV following Ang 1-7 treatment, but not in the control vehicle-treated group. Similar findings were described by Handa et al. where UF and UNaV were increased following intra-renal injection of both low and high doses of Ang 1-7 [37]. Discrepancies between the results in our study and the two above-mentioned studies could be attributed to variables in the experimental protocols, including differences in infusion technique, i.e., in situ vs. in vivo in DelliPizzi et al. [36] and intrarenal infusion in Handa et al. [37], different anesthetics, shorter equilibration and clearance periods and different Ang 1-7 infusion rates. For instance, Handa et al. used an infusion dose of Ang 1-7 that was 3 times higher than the two highest doses applied in our study [37]. Also, Dellipizzi et al. showed that infusion of Ang 1-7 alone does not change renal vascular resistance [36]. Similarly, van der Wouden et al. showed that perfusion of Ang 1-7 alone exerts no effect on the diameter of isolated small renal arteries, and the renal vascular resistance in isolated perfused kidneys remains unchanged [35]. However, Ang 1-7 was shown to completely inhibit the vasoconstrictor effect of Ang II in isolated small interlobar renal arteries [35]. Furthermore, Ang 1-7 diminishes the vasoconstrictor effect of Ang II in the efferent arterioles in vitro, but not in the afferent arterioles. However, in vivo infusion of Ang 1-7 in freely moving rats, it failed to attenuate the reduction of RBF in response to Ang II [35].

Our results show that urinary cGMP levels were higher in CHF rats as compared to sham controls. cGMP generation is upregulated in CHF probably due to activation of natriuretic peptide (ANP and BNP) and NO systems [38,39]. Since cGMP acts as a second messenger for Ang 1-7 [40,41], infusion of the latter into sham control and CHF rats resulted in further enhancement in urinary cGMP excretion [40]. These findings suggest that Ang 1-7 activates the renal NO/cGMP signaling cascade which may antagonize the deleterious effects of Ang II in CHF. The fact that urinary cGMP levels slightly decreased following AVE 0991 treatment of CHF group is unexpected and contradictory to a previous report published by Wiemer et al. [42], who showed that both Ang 1-7 and AVE 0991 increase NO release by cultured bovine endothelial cells. In addition, the duration of AVE 0991-mediated NO release was much longer than the duration of Ang 1-7-mediated NO release.

Finally, acute infusion of either Ang 1-7 or AVE0991 reduced MAP in sham-operated rats and to a lesser extent in CHF animals. These findings are in line with numerous other studies showing that Ang 1-7 and AVE 0991 are effective in lowering MAP under normal conditions and various cardiac diseases, via activation of NO/cGMP intracellular axis [35,41,43,44,45]. Since heart failure, especially during evolved stage, is characterized by low blood pressure/effective blood volume, further reduction of blood pressure is undesired. Such a side effect of certain therapies such as Nesiritide (recombinant BNP) may offset their beneficial action and limit their use for HF. However, since the depressor action of Ang 1-7 is not dramatic, the beneficial effects of this peptide justify its use despite its blood pressure-lowering properties.

### 3.2. Chronic Protocol

This protocol was designed to investigate the chronic effects of Ang 1-7 and its agonist AVE 0991 on renal function, cardiac hypertrophy and RAAS in experimental CHF. As expected, untreated CHF rats displayed lower cumulative UF, UNaV and UKV than their sham controls. These results are in line with our previous findings that rats with experimental ACF display avid sodium retention [29,46]. The mechanisms underlying this phenomenon include activation of neurohormonal systems such as RAAS, SNS, ADH and endothelin [9,10,11,29]. In addition, perturbations in the NO and natriuretic system may contribute to the exaggerated Na^+^ retention [47]. NO has been shown to inhibit NaCl reabsorption in the mTAL, thereby increasing NaCl excretion by the kidney [48]. In addition, Ang II exerts inhibitory effects on NO production through activation of AT1R in the mTAL in the kidney [48]. Furthermore, activation of the AT1R increases ROS generation and subsequently reduces NO production, whereas activation of the AT2R promotes NO production [48]. It should be emphasized that that CHF is associated with increased expression of Na^+^-K^+^-2Cl^−^ cotransporter in the mTAL, where ~25% of the NaCl filtered load are reabsorbed [48,49].

Notably, chronic administration of Ang 1-7 or AVE 0991 exerted significant diuretic, natriuretic and kaliuretic responses in CHF rats, but not in sham controls. In this context, Ang 1-7 and AVE 0991 resulted in steeper cumulative UF, UNaV and UKV compared to the summation curves observed in CHF vehicle-treated rats. Actually, the cumulative excretion curves of UF, UNaV and UKV in CHF animals chronically treated with Ang 1-7 or AVE 0079 were comparable to those observed in sham controls. These results indicate that Ang 1-7 and its agonist improve sodium balance in CHF rats, but not in healthy sham controls. The chronic effects of Ang 1-7 on both UNaV and UKV may stem from its stimulatory actions on renal hemodynamics or direct tubular effect [33,37,38]. This concept is supported by our findings that urinary cGMP levels in Ang 1-7- and AVE 0991-treated CHF rats were elevated compared to untreated CHF rats. It was reported that cGMP promotes degradation of cAMP, reduces Na^+^-K^+^-2Cl^−^ activity and ultimately increases electrolyte excretion [49], which corresponds to the current findings. The observed stimulatory effects of Ang 1-7 and AVE 0991 on urinary cGMP excretion are in agreement with those reported by Stegbauer et al. who demonstrated that chronic administration of Ang 1-7 in apoE-deficient mice resulted in a significant increase in cGMP levels in renal cortices and enhanced expression of eNOS mRNA in isolated preglomerular arteries [50]. In contrast to the higher basal UcGMP obtained in CHF as compared with sham controls of acute protocols, basal UcGMP in chronic protocol was comparable in vehicle-treated CHF rats and their sham controls. This surprising behavior could be attributed to the fact that daily urine collection was performed without cooling the tubes, whereas in the acute protocol urine samples were collected into vials kept in ice to prevent degradation of cyclic GMP. Most recently, Gawrys et al. demonstrated that administration of Vericiguat (BAY41-8543), a sGC stimulator, into hypertensive rats with experimental HF induced by the placement of A-V fistula, significantly improved survival rate of these rats along with increased ucGMP and decreased blood pressure [51]. Similar results were obtained in clinical trials where treatment with Vericiguat lowered the incidence of death from cardiovascular causes or hospitalization for heart failure among patients with heart failure and reduced ejection fraction [52]. These findings support the beneficial role of cGMP in heart failure and support the use of sGC stimulators as therapeutic approach for the treatment of the clinical entity, especially when a cardio–renal syndrome is evident.

Raffai et al. demonstrated that acute and chronic treatment with Ang 1-7 or AVE 0991 restored endothelium-dependent vascular relaxation in salt-fed Sprague–Dawley rats by reducing vascular oxidant stress and enhancing NO availability via Mas and AT2 receptors [43]. An additional system that potentially attributes to the stimulatory effects of Ang 1-7 on renal function is the SNS. In this regard, Torp et al. demonstrated that renal bilateral surgical denervation abolishes the increase in Na^+^-K^+^-2Cl^−^ cotransporter in the outer medulla of rats with experimental CHF [49]. This suggests that renal nerve has an impact on regulating Na^+^-K^+^-2Cl^−^ expression, and Ang 1-7- may suppress renal sympathetic nervous activity [53] and subsequently promotes excretion of urinary sodium and potassium in CHF.

As mentioned earlier, CHF is characterized by impaired renal perfusion, due to constriction of renal arteries secondary to low RBF. Prolonged renal hypoperfusion often results in reduction in kidney size. As anticipated, the KW/BW% values in all CHF groups were small compared to their sham controls. Treatment with Ang 1-7 and AVE 0991 in both CHF and sham control rats did not yield any significant effects on KW/BW%. sCr and aldosterone levels were significantly higher in vehicle-treated CHF rats as compared with their sham controls, and treatment with Ang 1-7 and AVE 0991 reduced these parameters to comparable levels as observed in sham controls. At four weeks of treatment, all groups exhibited lower sCr values than those observed with the vehicle group. Elevated sCr concentration reflects impaired kidney function in various clinical conditions, including CHF [54]. Therefore, it is not surprising that sCr levels were higher in vehicle-treated CHF rats than in vehicle-treated sham controls at two weeks after the induction of the disease. All CHF rats treated with Ang 1-7 and AVE 0991 had significantly lower sCr values than the vehicle-treated CHF 2W group. These findings correspond the stimulatory effects of Ang 1-7 and its mimetic on GFR and RPF in sham controls and to a lesser extent in CHF rats. At four weeks, the sCr levels were comparable in both CHF and sham controls treated with vehicle. Since SCr is produced and released by muscle turnover, the decrease in this parameter could be secondary to muscle loss, which often occurs in CHF [55].

Plasma BNP levels were significantly higher in CHF rats after 4 but not after 2 weeks from the induction of the disease as compared with sham controls. After 2 but not 4 weeks of treatment, plasma BNP levels in Ang 1-7-treated sham controls were much higher than those of vehicle-treated sham controls. Sham controls treated with AVE 0991 did not display significant change in plasma BNP levels. CHF rats treated with Ang 1-7 or AVE 0991 for 2 and more profoundly for 4 weeks exhibited slightly higher plasma BNP levels than those of vehicle-treated CHF rats. The stimulatory effect of both Ang 1-7 and AVE 0079 on BNP levels is in line with the reports that Ang 1-7 stimulates ANP secretion via MasR and PI3K-Akt-NOS pathway [56]. The lack of inhibitory effect of Ang 1-7 on BNP levels occurred despite the beneficial antihypertrophic actions of Ang 1-7 in CHF rats. These findings are in line with other studies which demonstrated beneficial cardioprotective effects of Ang 1-7 and its agonist AVE 0991 on cardiac hypertrophy and remodeling [57,58,59]. For instance, Patel et al. showed that chronic administration of Ang 1-7 reduces cardiac myocyte hypertrophy in ACE2 knockout mice subjected to pressure overload [57]. In contrast to our results, these authors demonstrated that mice chronically treated with Ang 1-7 had significantly lower cardiac BNP levels as compared with placebo-treated animals. Grobe et al. showed that co-infusion of Ang 1-7 with Ang II in Sprague–Dawley rats reduces the proliferative and hypertrophic effects of Ang II [58]. Ferreira et al. showed that AVE 0991 prevents cardiac hypertrophy induced by isoproterenol treatment in rats [59]. Flores-Munoz et al. demonstrated that delivery of adenoviral virus that overexpresses Ang 1-7 into cardiomyocytes attenuated hypertrophy induced by isoproterenol or Arg-vasopressin [60].

We also measured plasma aldosterone and Ang II levels after two weeks and four weeks from the initiation of treatment in CHF and sham control rats. As expected, the plasma levels of aldosterone in the CHF groups were higher after 4 weeks than those obtained in sham controls and even than after two weeks from the induction of the disease. The vehicle-treated CHF rats displayed ~two-fold increase in plasma aldosterone levels as compared with their sham controls. CHF rats treated with Ang 1-7 and AVE 0991 had much lower plasma aldosterone levels than those observed with the vehicle-treated CHF group. Specifically, CHF rats treated with Ang 1-7 and AVE 0991 showed plasma aldosterone concentrations comparable to their sham controls. Four weeks of Ang 1-7 treatment did not attenuate the elevated plasma aldosterone levels in CHF, whereas AVE 0079 significantly suppressed aldosterone secretion. There is accumulating evidence that high plasma aldosterone concentrations are linked to cardiac disease, including CHF [61]. In the current study and in another publication from our laboratory, we demonstrated that plasma levels of aldosterone increase two weeks and four weeks after the placement of ACF [29]. According to our findings, Ang 1-7 reduces plasma levels of aldosterone after 2 but not 4 weeks of treatment. These findings may indicate a cross-reaction between Ang 1-7 and aldosterone, as evident by the inhibitory effects of Ang 1-7 on aldosterone synthesis [61]. There is a substantial difference in aldosterone levels between Ang 1-7- and AVE 0991-treated CHF 4W subgroups. Since aldosterone is an important factor in cardiac fibrosis and may lead to profound differences in diastolic function between these two subgroups, the measurement of cardiac function using P–V loops or echocardiogram and the determination of the extent of myocardial fibrosis are appealing and should be addressed in future studies. In contrast to our assumption and extensive evidence from literature [61,62], the plasma levels of Ang II in CHF rats after two weeks and four weeks from the induction of CHF were comparable to those of sham control animals. For example, Luchner et al. demonstrated that plasma, renal, myocardial and vascular levels of Ang II are increased in dogs with experimental CHF [62]. The differences between our and their results could be attributed to different methods for measuring plasma Ang II. Specifically, Luchner et al. [62] used radioimmunoassay, whereas we used a commercial ELISA assay. Contaminants and interferences can be attributed to false results in the ELISA method. Radioimmunoassay is considered a highly sensitive technique; however, it requires special training and authorization due to the use of radioactive traces. Furthermore, Ang 1-7 and AVE 0991 treatment did not yield any significant inhibitory effects on plasma Ang II levels, despite the fact that both agents cause hypotension, which may activate RAAS. An alternative approach is to measure local levels of Ang II, such as within the kidneys rather than systemic Ang II, that may represent the real status of RAAS.

In conclusion, chronic administration of Ang 1-7 and its mimetic AVE 0991 exerts beneficial renal and cardiac effects in rats with CHF. Thus, the ACE2/Ang 1-7 axis represents a compensatory response to over-activity of the classic ACE/AngII/AT1R system, suggesting that Ang 1-7 may be a potential therapeutic agent in this clinical setting. Considering the local depletion of Ang 1-7 during SARS-CoV-2 infection, our findings support the use of Ang 1-7 to prevent heart, renal and lung damage during severe cases of COVID-19.

## 4. Materials and Methods

Studies were conducted on a local strain of male Sprague–Dawley rats (Harlan Laboratories, Ltd., Jerusalem, Israel), weighing 300–350 g. The animals were kept in individual metabolic cages in a temperature-controlled room and were fed standard rat chow (Altromin 1324 Standard Diet formula) containing 2156 mg/Kg sodium and 9214 mg/kg potassium and tap water ad libitum. All experiments were performed according to the guidelines of the committee for the supervision of animal experiments, Technion, IIT (IRB, IL0250212).

### 4.1. The Experimental Model

Heart failure was induced by surgical creation of an arterio-venous fistula between the abdominal aorta and the inferior vena cava by the method originally described by Stumpe et al. [63]. The abdominal aorta and inferior vena cava were exposed through a mid-abdominal incision under pentobarbital anesthesia (60 mg/kg, i.p.). Miniature surgical clamps were placed around both vessels, 10–15 mm apart, and a longitudinal incision was performed in the outer wall of the vena cava. The common wall between the aorta and the vena cava was grasped through incision under binocular magnification, and a fistula (1.2 mm OD) was created between the two vessels. The opening of the outer wall of the vena cava was then closed with a continuous suture (7-0 prolene nonabsorbable suture, Ethicon) [8,9,10,11]. After the surgical procedure, the animals were allowed to recover and then returned to the metabolic cages for daily monitoring of urine output and sodium excretion. Acute CHF was considered after 7 days of monitoring while chronic CHF was defined as 28 days after ACF induction. A-V fistula-induced heart failure model exhibits several manifestations of this clinical setting including kidney dysfunction, neurohormonal activation and cardiac hypertrophy [29]. After 2 and 4 weeks from the induction of the disease, rats are still in the compensatory phase and considered HFpEF despite few typical characteristics of heart failure such as activation of RASS, SNS, ADH, ANP/BNP, impaired kidney function and cardiac remodeling. It would take 8 weeks until the animals convert to HFrEF.

### 4.2. Acute Studies

CHF rats (*n* = 7–10) and control rats (*n* = 9–12) were anesthetized with Inactin (thiobutabarbital sodium, 100 mg/kg i.p.; Sigma Chemicals, St. Louis, MO, USA) and operated on a thermos-regulated surgical table to maintain their body temperature at 37 °C. Initially, tracheostomy was performed and followed by insertion of polyethylene catheters (PE50, Portex Ltd., England, UK) into the left carotid artery, jugular vein and urinary bladder for monitoring mean arterial pressure (MAP), infusion of different solutions and collection of urine and blood samples, respectively. Mean arterial pressure was measured through the carotid arterial line using a pressure transducer (model 156PC05GWL; Microswitch, Freepoint, IL, USA). A solution of inulin (2%)+PAH (0.5%) was infused intravenously at a rate of 1.5 mL per hour throughout the experiments. After surgery and 60 min of equilibration, two baseline periods of 30 min each were collected. Afterwards, the rats were infused with increasing doses of Ang 1-7 (0.3, 3, 30 and 300 ng/kg/min) or AVE 0991 (0.67, 2.7 and 13.8 µmole/kg/h). Rats that were treated with vehicle (saline) served as controls. A 30 min clearance period was obtained with each dose. Urine samples were collected into pre-weighed vials containing mineral oil and kept in ice in order to prevent degradation of cyclic GMP (cGMP). Blood samples were collected (0.3 mL) into heparinized tubes at the midpoint of each clearance period and centrifuged to separate blood from plasma. Plasma and urine samples were kept in (−20 °C) for analyzing GFR, RPF, electrolytes and cGMP as described below.

### 4.3. Chronic Studies

This protocol was designed to investigate the effects of long-term administration of Ang 1-7 or AVE 0991 on kidney function and cardiac hypertrophy in CHF as compared with normal animals (*n* = 6). Animals of all groups were kept in individual metabolic cages for daily measurements of urine volume and urinary sodium and potassium excretion at baseline periods (5 days) and for an additional 4 weeks of treatment. The animals received a solution of either Ang 1-7 or AVE 0991 (24 µg/kg/h, i.p.) via intra-peritoneally implanted osmotic minipumps (alzet 2004). Blood samples were withdrawn from the tail after two weeks. At the end of the treatment period, the animals were sacrificed, blood samples collected and their hearts and kidneys were removed. Heart weights were measured to assess the effects of these compounds on the progression of cardiac remodeling in CHF.

### 4.4. Physiological and Chemical Analyzes

**Urine flow:** Urine flow (UF) was calculated by dividing urine volume to time of urine collection; **GFR and RPF:** Concentrations of inulin in urine and plasma samples were determined using the colorimetric anthrone method. **Glomerular filtration (GFR)** was calculated by the following equation: (Urine inulin) × (Urinary flow rate)/(Plasma inulin). **Renal plasma flow (RPF)** was estimated by measuring concentrations of PAH in plasma and urine samples. RPF was calculated in a similar way to GFR: (Urine PAH) × (Urinary flow rate)/(Plasma PAH); **Sodium excretion**: Sodium concentrations in the urine and plasma were measured using a flame photometer (model IL 943, Instrumentational Laboratory). **Urinary sodium excretion (UNaV)** was calculated by UNa × Urinary flow rate; **Fractional excretion of sodium (FENa)** was determined by the following equation: UNaxV/(*PNa × GFR)%; *UNa—Urinary sodium concentration; *PNa—Sodium concentration in plasma; The cumulative daily urine volume, urinary sodium and potassium excretion was obtained from the additive daily value from day 0 up to 28 days of treatment period. **Urinary cyclic GMP (cGMP):** Since cGMP is the major second messenger of Ang 1-7, urinary excretion of cGMP in the various experimental groups was determined using Cyclic GMP EIA kit (Cayman Chemical); **Serum creatinine levels:** Serum creatinine (sCr) levels were measured in blood samples withdrawn from the tail after two weeks and on the last day of the chronic treatment period using a commercialized EIA kit (Cayman Chemical); **Plasma BNP:** Plasma BNP levels were determined in blood samples using an EIA kit purchased from Assaypro LLC; **RAAS status**: Ang II levels were determined in plasma samples collected in EDTA tubes using an EIA kit purchased from Assaypro. Plasma aldosterone levels were measured with EIA kit purchased from Cayman Chemical.

### 4.5. Statistical Analysis

Data are expressed as mean ± SEM and presented as graphs drawn by GraphPad prism 5.0 software. One-way ANOVA followed by Tukey test was applied within group comparisons in the acute studies. In the chronic studies, two-way ANOVA followed by Bonferroni post-tests were used to compare between treated and untreated animal groups and replicate means were calculated at different time points.

## Figures and Tables

**Figure 1 ijms-24-11470-f001:**
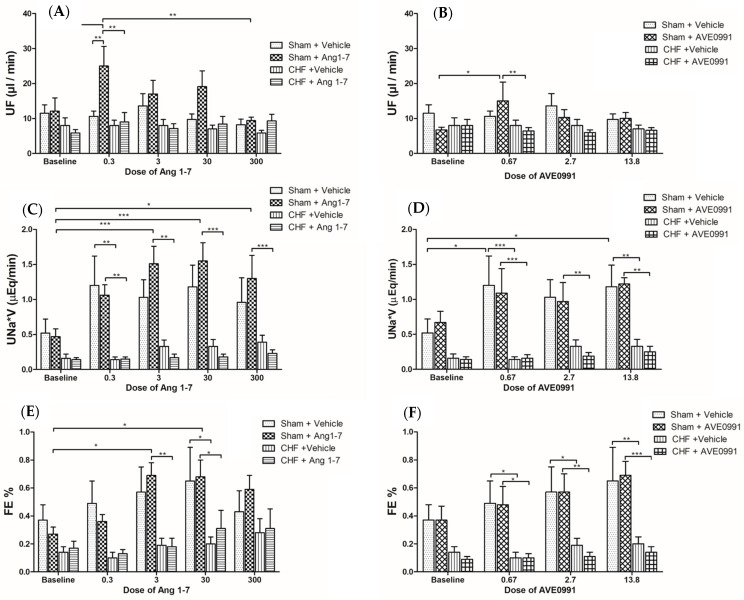
Acute changes from baseline in urinary flow (UF), urinary sodium excretion (UNaV) and fractional sodium excretion (FENa) following acute infusion of increasing doses of Ang 1-7 (**A**,**C**,**E**) and AVE 0991 (**B**,**D**,**F**) in control or CHF rats. One-way ANOVA followed by Tukey test was applied within group comparisons in the acute studies. * *p* < 0.05, ** *p* < 0.01, *** *p* < 0.001 vs. baseline values.

**Figure 2 ijms-24-11470-f002:**
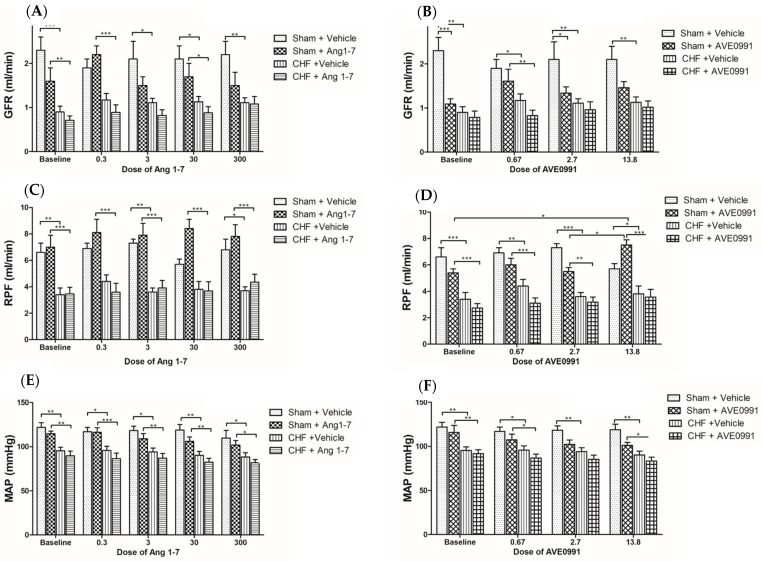
Acute changes in glomerular filtration rate (GFR), renal plasma flow (RPF) and mean arterial blood pressure (MAP) following acute infusion of increasing doses of Ang 1-7 (**A**,**C**,**E**) and AVE 0991 (**B**,**D**,**F**) in control or CHF rats. One-way ANOVA followed by Tukey test was applied within group comparisons in the acute studies. * *p* < 0.05, ** *p* < 0.01, *** *p* < 0.001 vs. baseline values.

**Figure 3 ijms-24-11470-f003:**
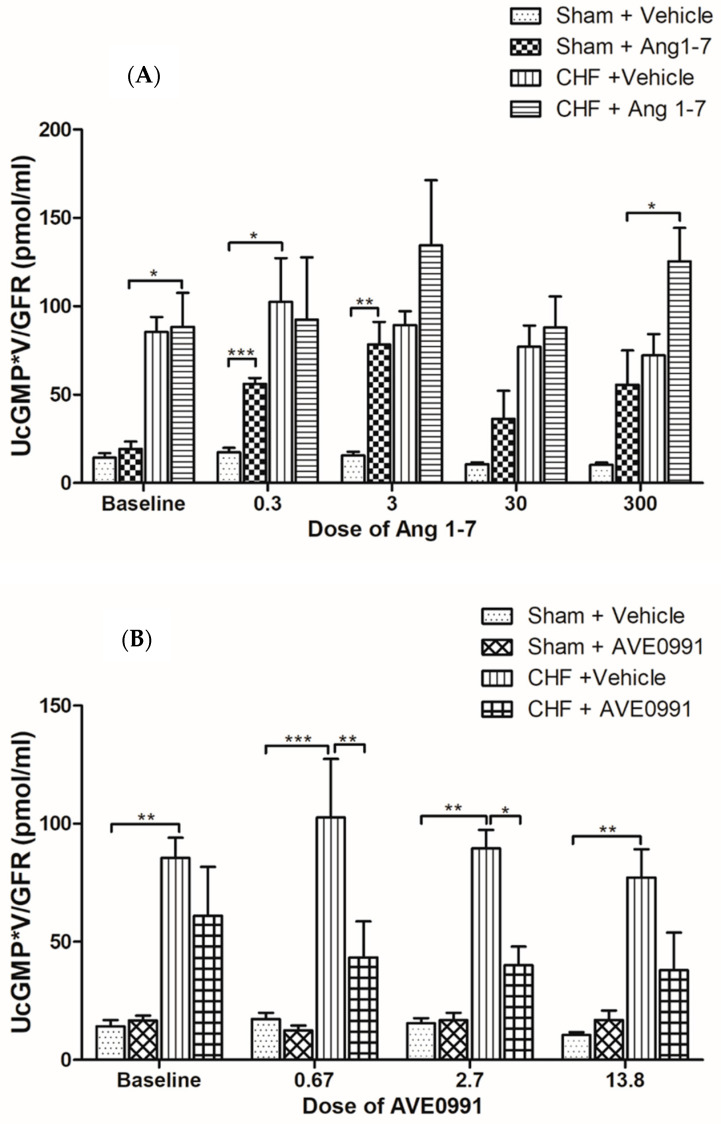
Changes in urinary excretion of cGMP normalized to GFR in CHF and control rats following acute treatment with incremental doses of Ang 1-7 (**A**) or AVE 0991 (**B**). One-way ANOVA followed by Tukey test was applied within group comparisons in the acute studies. * *p* < 0.05, ** *p* < 0.01, *** *p* < 0.001 CHF vs. sham controls, or treated CHF and sham vs. corresponding untreated subgroups.

**Figure 4 ijms-24-11470-f004:**
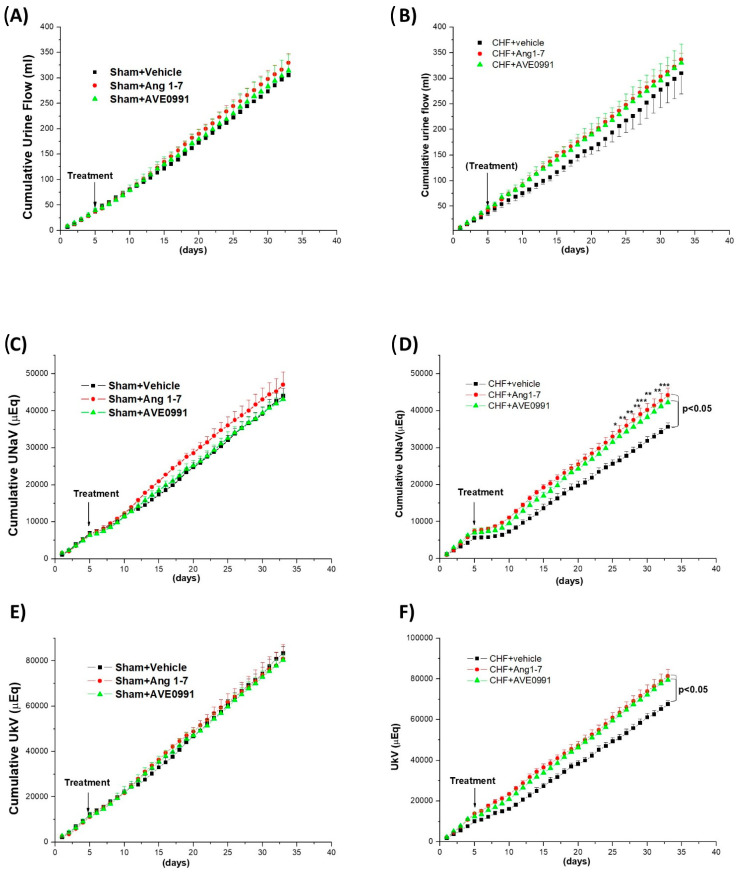
Effects of chronic treatment with Ang 1-7 and its mimetic (AVE 0991) on cumulative urinary flow (UF), Na^+^ excretion (U_Na_V) and K+ excretion (U_K_V) in control rats (**A**,**C**,**E**) and CHF animals (**B**,**D**,**F**). Cumulative urine volume (**A**,**B**), sodium excretion (**C**,**D**) and potassium excretion (**E**,**F**) in control rats and in CHF rats chronically treated with Ang 1-7 or AVE 0079 at a dose of 24 µg/kg/h i.p. or vehicle via osmotic minipumps for 28 days. Baseline values refer to 5-day collection periods (days 1–5) prior to initiation of the various treatment. * *p* < 0.05, ** *p* < 0.01, *** *p* < 0.001 compared with untreated CHF. Two-way ANOVA followed with Bonferroni post-tests was used to compare between treated and untreated animal groups.

**Figure 5 ijms-24-11470-f005:**
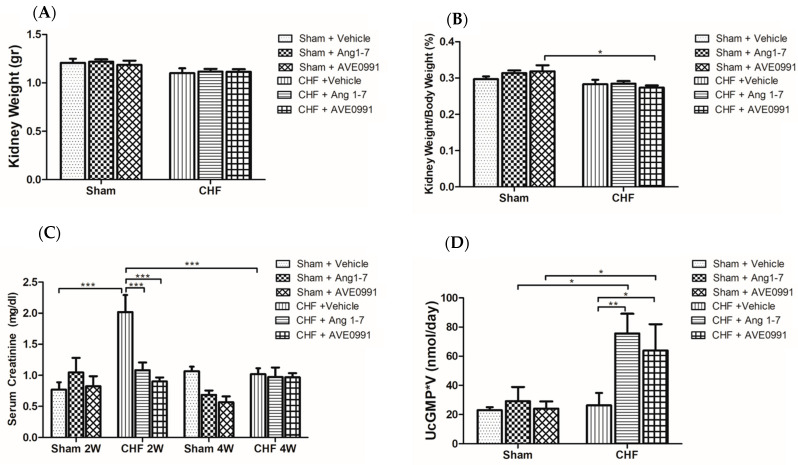
Effects of chronic treatment with Ang 1-7 and its mimetic (AVE 0991) on absolute kidney weight (KW) (**A**), normalized kidney weight to body weight (KW/BW%) (**B**), serum creatinine (sCr) (**C**) and daily urinary cGMP excretion (UcGMP*V) (**D**) after 2 and 4 weeks of treatment of control rats and CHF animals. One-way ANOVA followed by Tukey test was applied within group comparisons in the acute studies. * *p* < 0.05, ** *p* < 0.01, *** *p* < 0.001 between CHF and their sham controls or treated and untreated CHF animals or sham controls.

**Figure 6 ijms-24-11470-f006:**
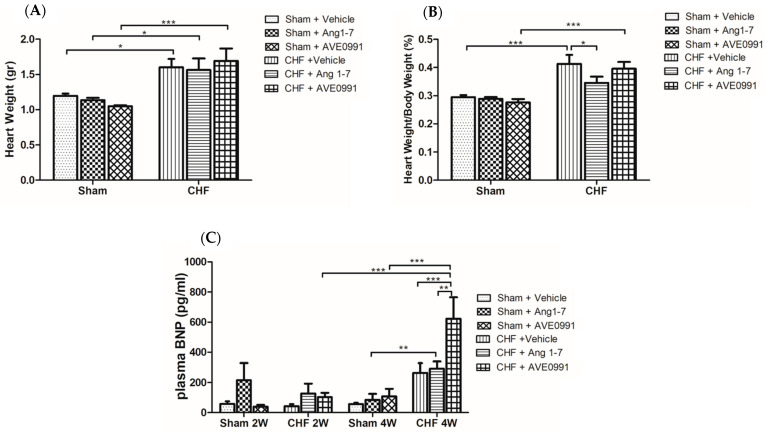
Effects of chronic treatment with Ang 1-7 and its mimetic (AV E0991) on absolute heart weight (HW) (**A**), normalized heart weight to body weight (HW/BW) (**B**) after 4 weeks of treatment and plasma BNP levels (**C**), after 2 and 4 weeks of treatment of control rats and CHF animals. One-way ANOVA followed by Tukey test was applied within group comparisons in the acute studies. * *p* < 0.05, ** *p* < 0.01, *** *p* < 0.001 between CHF and their sham controls or treated and untreated CHF animals or sham controls.

**Figure 7 ijms-24-11470-f007:**
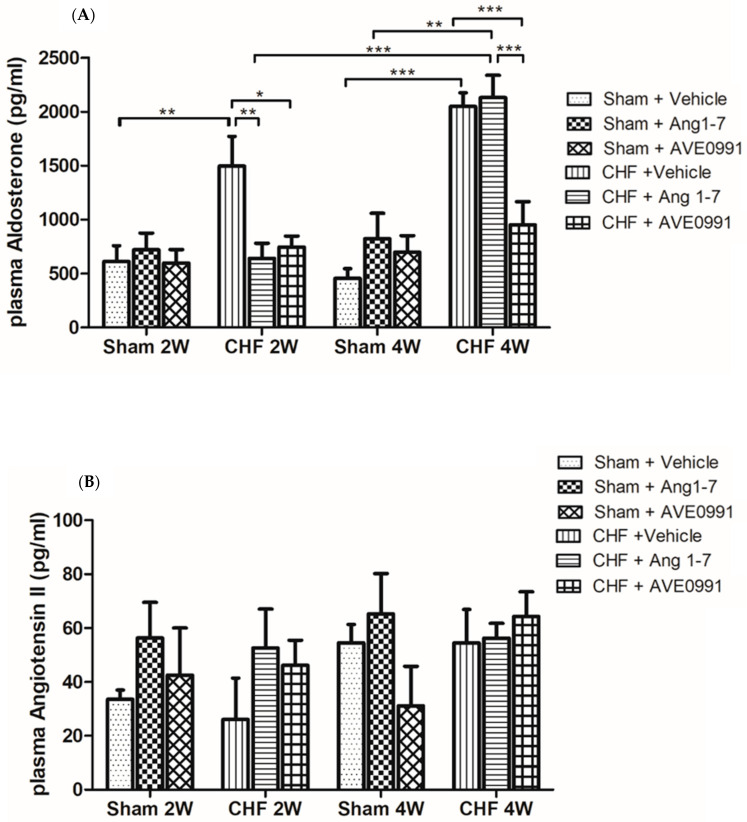
Effects of chronic treatment with Ang 1-7 and its mimetic (AVE 0991) on plasma aldosterone (**A**) and angiotensin II (**B**) levels after 2 and 4 weeks of treatment of control rats and CHF animals. One-way ANOVA followed by Tukey test was applied within group comparisons in the acute studies. * *p* < 0.05, ** *p* < 0.01, *** *p* < 0.001 between CHF and their sham controls or treated and untreated CHF animals or sham controls.

## Data Availability

The data presented in this study are available upon request from the corresponding author.

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
