# Peer review of "Effects of Angiotensin 1-7 and Mas Receptor Agonist on Renal System in a Rat Model of Heart Failure"

_ijms, 2023, doi:10.3390/ijms241411470_

Round 1

Reviewer 1 Report

Comments and Suggestions for Authors

Dear editor, carefully review the article with title “Renal and Cardiac Effects of Angiotensin 1-7 and Mas Receptor Agonist in a Rat Experimental Heart Failure# by the authors Ravit Cohen-Segev, Omri Nativ , Safa Kinaneh, Doron Aronson, Aviva Kabala, Shadi Hamoud, Tony Karram, Zaid Abassi, Where the authors  postulate that ACE2/Ang 1-7 axis represents a compensatory response to over-activity of ACE/AngII/AT1R system characterizing CHF and suggests that Ang 1-7 may be a potential therapeutic agent in this disease state.

I have a number of questions about this manuscript.

The title could be changed to “Effects of angiotensin 1-7 and Mas receptor agonist on of renal system in a rat model the heart failure”

Page 2, line 29 please deleted and their sham controls

Please added in one or two phrase the methods

Page 2, lines 32, 34. Please define the abbreviations UNaV, GRF,  sCr,  RPF and cGMP

Please explain the effect of AVE0991 agonist

Substitute the symbol & in all manuscript, for and this symbol is not used in scientific papers

Please, substitute in the histograms a different color for each bar because as they are there is confusion, Figure 1 and 2 (A, B, C, D, E, and F)

Change the abbreviation of "V" (urinary flow) to UF

Since the authors have the hearts and kidneys of the experimental animals, could you please show the histology of the structural changes that these organs present (which according to the results should present these changes?)

Vehicle-treated rats should show no change in the parameters that were tested. However, if there are any, how the authors explain these changes.

Because the experimental groups are presented in figures 1 and 2 in a different way than in figures 4, 5, 6 and 7, they could standardize the presentation of the results, figures 1 and 2 are very confusing.

Page 20 lines 255-267, as this section does not seem like a discussion, but a description of results, please could you modify it or change it to the results section

In the methods section the authors never explain what was in the vehicle so they could add it

The methods section does not show how the experimental groups were formed or the number of animals per group.

Please add the composition of the rat’s food

Please add the protocol approval number

Because two different anesthetics were used, as well as the dose please justify this.

Please change healthy controls to rats control throughout the manuscript

Author Response

We acknowledge the reviewer for his/her valuable comments. Please find below a point-to-point reply to these comments:

Reviewer I

Dear editor, carefully review the article with title “Renal and Cardiac Effects of Angiotensin 1-7 and Mas Receptor Agonist in a Rat Experimental Heart Failure# by the authors Ravit Cohen-Segev, Omri Nativ , Safa Kinaneh, Doron Aronson, Aviva Kabala, Shadi Hamoud, Tony Karram, Zaid Abassi, Where the authors  postulate that ACE2/Ang 1-7 axis represents a compensatory response to over-activity of ACE/AngII/AT1R system characterizing CHF and suggests that Ang 1-7 may be a potential therapeutic agent in this disease state.

I have a number of questions about this manuscript.

The title could be changed to “Effects of angiotensin 1-7 and Mas receptor agonist on of renal system in a rat model the heart failure” – Done, The suggested title was adopted, (Page 1).

Page 2, line 29 please deleted and their sham controls - Done

Please added in one or two phrase the methods- Done

Page 2, lines 32, 34. Please define the abbreviations UNaV, GRF,  sCr,  RPF and cGMP- Done, (Page 2)

Please explain the effect of AVE0991 agonist-

We refereed to the renal and cardiac effects of AVE0991 in the abstract, results and discussion.

Substitute the symbol & in all manuscript, for and this symbol is not used in scientific papers- Done, & symbol was omitted throughout the MS.

Please, substitute in the histograms a different color for each bar because as they are there is confusion, Figure 1 and 2 (A, B, C, D, E, and F).

The histograms of the various experimental groups are expressed in different pattern throughout the attached figures, in black and white rather than using different colors.

Change the abbreviation of "V" (urinary flow) to UF- Done “V” was changed to “UF” throughout the MS.

Since the authors have the hearts and kidneys of the experimental animals, could you please show the histology of the structural changes that these organs present (which according to the results should present these changes?)-

Thank you for this important issue. Unfortunately, we did not perform histological analysis of the hearts and kidneys. Although, we do not expect to see any cardiac and renal histological changes in the acute protocol, it may be of relevance to the chronic studies where histological alterations at the heart and kidney levels may be detected.

Vehicle-treated rats should show no change in the parameters that were tested. However, if there are any, how the authors explain these changes.

It is well known that administration of the vehicle is accompanied by minor volume addition that can result into slight changes of the studied parameters.

Because the experimental groups are presented in figures 1 and 2 in a different way than in figures 4, 5, 6 and 7, they could standardize the presentation of the results; figures 1 and 2 are very confusing.

The reason that we presented the results in Fig 1 and 2 in different pattern from figures 4-7, is the fact the we applied two experimental protocols (acute and chronic), where Fig 1 and 2 depict the acute phase whereas 4-7 summarize the chronic protocols. In order to avoid any confusion we marked the word acute vs chronic in the figure legends.

Page 20 lines 255-267, as this section does not seem like a discussion, but a description of results, please could you modify it or change it to the results section.

It is common to start the discussion with a paragraph which summarizes the major findings of the study. This paragraph was slightly rephrased to reflect discussion manner rather than descriptive findings. The real discussion comes immediately after this summarizing paragraph, where it was divided into acute and chronic protocols (Pages 11-20).

In the methods section the authors never explain what was in the vehicle so they could add it. We used saline as vehicle- We added this information to the methods section (Page 22, 1st paragraph).

The methods section does not show how the experimental groups were formed or the number of animals per group.

The number of animals in each group is provided in the method section (pages 21-22).

Please add the composition of the rat’s food

The composition of the chow is now provided (Methods, Page 20).

Please add the protocol approval number

The IRB number is now provided (Page 20)

Because two different anesthetics were used, as well as the dose please justify this.

Since we applied two different protocols, namely acute vs chronic, the duration of each protocol requires an appropriate anesthetic. Specifically, acute studies lasted for several hours, therefore we applied Inactin which is widely used in such lengthy acute in vivo experiments. In contrast, we applied Nembutal when we induced heart failure in rats for the chronic studies, a surgical procedure that takes about 30 min. 

Please change healthy controls to rats control throughout the manuscript.

“Healthy controls” was changed to “control rats” throughout the MS.

Reviewer 2 Report

Comments and Suggestions for Authors

Reviewing the manuscript entitled, “Renal and Cardiac Effects of Angiotensin 1-7 and Mas Receptor Agonist in a Rat  Experimental Heart Failure” by Cohen-Segev R et al., this focuses on potential mechanisms of Ang1-7 to MAS receptor on the heart failure with kidney disease in the acute and chronic phases. Although this is an interesting manuscript, the authors should check the basic conditions of the experiments. So, the authors need to respond to the following concerns.

 What is new evidence of this manuscript? Ang1-7 is known to act as an antithesis to AngII-AT1, although no consensus has been reached.

 As mentioned in material and methods, for making of experimental congestive heart, you used the aortic-venous fistula method. The mechanism that evokes heart failure is that renal blood flow is stolen and renin is activated, resulting in increased blood pressure and the formation of pressure overload leading to congestion. Is this correct? In other words, I think that cardiac hypertrophy does not occur unless there is some form of pressure overload on the heart. Why does CHF occur despite significantly lower MAP in the acute phase? Does doubling urine output affect blood pressure? Since he is in heart failure, I think renin is intensely activated. This mechanism is inevitable, at least because RAS activity is the compensation for life support during the acute phase of heart failure.

The authors mentioned Finally, acute infusion of either Ang 1-7 or AVE0991 reduced MAP in sham operated rats and to a lesser extent in CHF animals. These findings are in line with numerous other studies showing that Ang 1-7 and AVE0991 are effective in lowering MAP under normal conditions and various cardiac diseases, via activation of NO/cGMP intracellular axis [35, 41, 43-45]. Does this mean that administration of Ang 1-7 in the acute phase is undesirable?

 The authors should add results of cardiac function during acute and chronic heart failure. I think the degree of heart failure is the factor that most influences the outcome.

 Is there any change in the expression level of MAS receptor during heart failure?

 Figure 4 and its explanation in the text do not match. You mentioned “CHF rats treated with AVE0991 displayed similar trend, yet the 158 magnitude of the change was to a lesser extent (Fig. 4D).” at the line 158. This description is not good. You should state whether or not there is a statistically significant difference.

 In Figure 5D, the authors need to add the results of 2w in CHF group.

 In Figure 6, the authors need to add the data of the cardiac function such as cardiac echo, and then interpretate the results.

 In Figure 7, in spite of there is increased aldosterone level in Vehicle and Ang 1-7 of CHF 4W, why is there no difference in angiotensin II in all group in CHF 4W.

 In Figure 7, there is big difference of aldosterone level between Ang 1-7 and AVE0991 in CHF 4W. Aldosterone is an important factor in cardiac fibrosis, and this difference likely produces a large difference in diastolic function between the two. The authors should add the data that supports these results.

 The authors need to modify experimental number.

Comments on the Quality of English Language

There is no particular problem with manuscript.

Author Response

We acknowledge both reviewers for his/her valuable comments. Please find below a point-to-point reply to these comments:

Reviewer II

Comments and Suggestions for Authors

Reviewing the manuscript entitled, “Renal and Cardiac Effects of Angiotensin 1-7 and Mas Receptor Agonist in a Rat  Experimental Heart Failure” by Cohen-Segev R et al., this focuses on potential mechanisms of Ang1-7 to MAS receptor on the heart failure with kidney disease in the acute and chronic phases. Although this is an interesting manuscript, the authors should check the basic conditions of the experiments. So, the authors need to respond to the following concerns.

Comment 1: What is new evidence of this manuscript? Ang1-7 is known to act as an antithesis to AngII-AT1, although no consensus has been reached.

Response: As the reviewer mentioned Ang1-7/MasR axis is antithesis to Ang II/AT1 one. Although most studies support this concept, this matter is still not settled in many aspects, including the role of ang 1-7 in the pathogenesis of CHF and its potential therapeutic utilization in this clinical setting.  The current study suggests that ACE2/Ang-(1-7)/MasR axis plays a cardioprotective role in heart failure, where it attenuates the progression of cardiac hypertrophy and improves cardio-renal manifestations. This experimental research indicates that the stimulation of the ACE2/Ang-(1-7)/MasR axis may serve as a novel therapeutic option in heart failure entity. 

 Comment 2: As mentioned in material and methods, for making of experimental congestive heart, you used the aortic-venous fistula method. The mechanism that evokes heart failure is that renal blood flow is stolen and renin is activated, resulting in increased blood pressure and the formation of pressure overload leading to congestion. Is this correct? In other words, I think that cardiac hypertrophy does not occur unless there is some form of pressure overload on the heart. Why does CHF occur despite significantly lower MAP in the acute phase? Does doubling urine output affect blood pressure? Since he is in heart failure, I think renin is intensely activated. This mechanism is inevitable, at least because RAS activity is the compensation for life support during the acute phase of heart failure.

Response: The precise description of the reviewer is correct. Specifically, in the current research we applied an experimental model of heart failure induced by the placement of an aorto-caval fistula (A-V Fistula). We have used this model for more than 30 years to study the manifestations of CHF at the cardiac, renal and endocrine levels. Although the creation of the A-V fistula reduces MAP, it is associated with high venous return and eventually enhanced volume/pressure overload on the heart chambers. This behavior along activation of the RAAS system plays a major role in cardiac hypertrophy/remodeling and kidney dysfunction as we have demonstrated in numerous studies (See Refs. 8-11, 29).  Doubling urine output is provoked by the activation of natriuretic peptides system and probably contributes to the observed low blood pressure. Furthermore, we demonstrated in previous studies that rats with A-V fistula exhibit activation of renin in correlation with the severity of heart failure as a compensatory mechanism at early stage of the disease.

Comment 3: The authors mentioned “Finally, acute infusion of either Ang 1-7 or AVE0991 reduced MAP in sham operated rats and to a lesser extent in CHF animals. These findings are in line with numerous other studies showing that Ang 1-7 and AVE0991 are effective in lowering MAP under normal conditions and various cardiac diseases, via activation of NO/cGMP intracellular axis [35, 41, 43-45].” Does this mean that administration of Ang 1-7 in the acute phase is undesirable?

Response: Since heart failure, especially the evolved stage is characterized by low blood pressure and effective blood volume, reduction of blood pressure is undesired. This effect may offset the beneficial action of certain therapies such as Nesiritide (recombinant BNP) and limit its use for HF. Yet, if the reduction in MAP is not dramatic, the beneficial effects of such drugs overcome this side effect and their use is not seriously hampered. Since the depressor action of Ang 1-7 is modest, we do not exclude its utilization in chronic heart failure. We referred to this point in the revised MS (page 14).

Comment 4: The authors should add results of cardiac function during acute and chronic heart failure. I think the degree of heart failure is the factor that most influences the outcome.

Response: Indeed, cardiac function is an issue of special interest. Technical problems precluded us from performing cardiac function determination. The clearance studies that we applied to assess the acute effects of either Ang 1-7 or AVE0991 require canulation of jugular vein and carotid artery along urinary bladder. The cannulation of the carotid artery is required for monitoring blood pressure. To measure cardiac function via P-V loops we need to insert Millar Tip into the carotid artery. Cannulation of both left and right carotid arteries is fatal. Therefore, we chose to measure blood pressure rather than CO. Measuring CO requires duplicating the study if we choose to add additional group of animals. We wish for your understanding.

 Comment 5: Is there any change in the expression level of MAS receptor during heart failure?

Response: Thank you for calling our attention to this important issue. Unfortunately, we did not measure cardiac or renal MasR in the current study.

 Comment 6: Figure 4 and its explanation in the text do not match. You mentioned “CHF rats treated with AVE0991 displayed similar trend, yet the 158 magnitude of the change was to a lesser extent (Fig. 4D).” at the line 158. This description is not good. You should state whether or not there is a statistically significant difference.

Response: Thank you for the note. This statement was rephrased to reflect the results depicted in Fig. 4: Specifically, we included the following paragraph “cumulative UNaV values of AVE0079-treated CHF rats were significantly higher than those obtained in CHF rats treated with vehicle alone and reached to comparable levels of UNaV measured in sham controls” (Page 8, 1st paragraph).

 Comment 7: In Figure 5D, the authors need to add the results of 2w in CHF group.

Response: Unfortunately, we did not measure urinary cGMP in the urine samples collected from 2W in CHF group.

Comment 8: In Figure 6, the authors need to add the data of the cardiac function such as cardiac echo, and then interpretate the results.

Response: Unfortunately, we did not perform cardiac echo analysis, which indeed could be an appropriate approach to determine cardiac function and geometry, as we do not have echocardiogram scan in our lab and purchasing such device is beyond our financial abilities.

 Comment 9: In Figure 7, in spite of there is increased aldosterone level in Vehicle and Ang 1-7 of CHF 4W, why is there no difference in angiotensin II in all group in CHF 4W.

Response: Thank you for the note. Indeed, the aldosterone levels were high in CHF 4W rats, but Ang II were not elevated in these animals. As compared with the long half-life of aldosterone, Ang II half life is 30 sec which make accurate measurements unachievable.  It is well known issue that measurement of Ang II is challenging and requires novel approaches, like radioimmunoassay rather than ELIZA. We referred to this matter in the discusion (Page 20). 

 Comment 10: In Figure 7, there is big difference of aldosterone level between Ang 1-7 and AVE0991 in CHF 4W. Aldosterone is an important factor in cardiac fibrosis, and this difference likely produces a large difference in diastolic function between the two. The authors should add the data that supports these results.

Response: Thank you for this note. As we did not measure cardiac function neither systolic nor diastolic by P-V Loops or Echocardiogram, we could not speculate whether these differences in aldosterone have cardiac consequences in terms of diastolic function or cardiac fibrosis. However, we referred to this issue and the limitation of the current study where we did not measure cardiac function and fibrosis and the potential involvement of aldosterone in this phenomenon (Page 19).

 Comment 11: The authors need to modify experimental number.

Response: Done

Reviewer 3 Report

Comments and Suggestions for Authors

   In the article entitled “Renal and Cardiac Effects of Angiotensin 1-7 and Mas Receptor Agonist in a Rat Experimental Heart Failure”, the authors evaluated the impact of Ang1-7 and AVE0991 treatment on functional renal and cardiac parameters in rats with heart failure. The study is sound and contributes to knowledge in the field. To strengthen the manuscript, please address the following comments.

Major comments

1)    It would be interesting to add the histology of the heart to support the data regarding the section “2.2.1 Effect of Ang 1-7 and AVE0991 on Cardiac Remodeling”.

2)    In the section “2.2.3 Effect of Ang 1-7 and AVE0991 on RAAS Status”, the authors observed that there was no difference in systemic levels of plasma Ang II, but are there any results concerning local levels, such as within the kidneys?

3)    Is the analysis of renal histology available in control and Ang1-7 and AVE 0991-tretaed animals? Was acute tubular necrosis found or was the renal histology normal when the groups were compared?     

4)    For the sake of transparency, all bar charts should be changed to bar chart plus scatter chart format. Please indicate the number of animals used in each experiment.  

Minor comments

1) Page 4, line 70, lacks reference.

2) Figures 1A and B: I recommend setting the same scale for both graphs.

3) In figure 4, its legend contains the applied statistical test. I also suggest indicating the tests used in the other figures

Author Response

We acknowledge the reviewer for his/her valuable comments. Please find below a point-to-point reply to these comments:

Reviewer III

In the article entitled “Renal and Cardiac Effects of Angiotensin 1-7 and Mas Receptor Agonist in a Rat Experimental Heart Failure”, the authors evaluated the impact of Ang1-7 and AVE0991 treatment on functional renal and cardiac parameters in rats with heart failure. The study is sound and contributes to knowledge in the field. To strengthen the manuscript, please address the following comments.

Major comments

Comment 1:    It would be interesting to add the histology of the heart to support the data regarding the section “2.2.1 Effect of Ang 1-7 and AVE0991 on Cardiac Remodeling”.

Response: Thank you for this important issue. Unfortunately, we did not perform histological analysis of the hearts and kidneys. Although, we do not expect to see any cardiac and renal histological changes in the acute protocol, it may be of relevance to the chronic studies where histological alterations at the heart and kidney levels may be detected. As seen in the MS we determined heart weight and HW/BW ration, which clearly show cardiac hypertrophy, a hallmark feature of cardiac remodeling. We agree that determining cardiac fibrosis may support or conclusion.

Comment 2;    In the section “2.2.3 Effect of Ang 1-7 and AVE0991 on RAAS Status”, the authors observed that there was no difference in systemic levels of plasma Ang II, but are there any results concerning local levels, such as within the kidneys?

Response: Thank you for the note. Indeed, the aldosterone levels were high in CHF 4W rats, but circulating Ang II levels were not elevated in these animals. As compared with the long half-life of aldosterone, Ang II half-life is 30 sec which make accurate measurements unachievable.  It is well known issue that measurement of Ang II is challenging and requires novel approaches, like radioimmunoassay rather than ELIZA. Your suggestion to meaure Ang II in the kidney is interesting, but as mentioned measurement of Ang II in the plasma and even locally in the kidney with ELIZA is inaccurate.  We referred to this matter in the discussion (Page 20). 

Comment 3: Is the analysis of renal histology available in control and Ang1-7 and AVE 0991-tretaed animals? Was acute tubular necrosis found or was the renal histology normal when the groups were compared?   

Response: Thank you for this important issue. Unfortunately, we did not perform histological analysis of the kidneys. Although, we do not expect to see any renal histological changes in the acute protocol, it may be of relevance to the chronic studies where Ang 1-7 may ameliorate the adverse histological alterations.

Comment 4:    For the sake of transparency, all bar charts should be changed to bar chart plus scatter chart format. Please indicate the number of animals used in each experiment.

Response: The animal number in the various protocols is indicated (Page 21-22).

Minor comments

  • Page 4, line 70, lacks reference. Added (page 4).

2) Figures 1A and B: I recommend setting the same scale for both graphs. Done

3) In figure 4, its legend contains the applied statistical test. I also suggest indicating the tests used in the other figures. Done

Round 2

Reviewer 1 Report

Comments and Suggestions for Authors

Dear Authors and editor.

Thanks you for answers that you gave me and by attend to the suggestions that I made. No more questions and doubts, now your manuscript could be publisher in IJMS.

Best regards

The reviewer

Author Response

Thank you for your kind answer.

All your notes were addressed.

Language has been upgraded, as suggested

Reviewer 2 Report

Comments and Suggestions for Authors

Reviewing the manuscript entitled, “Renal and Cardiac Effects of Angiotensin 1-7 and Mas Receptor Agonist in a Rat Experimental Heart Failure” by Cohen-Segev R et al., this focuses on potential mechanisms of Ang1-7 to MAS receptor on the heart failure with kidney disease in the acute and chronic phases. The authors well responded to my concerns. So, the authors need to respond to the following concerns for acceptable quality.

Does this heart failure model exhibit HFrEF or HFpEF in the chronic phase? The authors should describe either HFrEF or HFpEF and its evidence in the manuscript.

Comments on the Quality of English Language

No particular problems were found.

Author Response

Comment: Does this heart failure model exhibit HFrEF or HFpEF in the chronic phase? The authors should describe either HFrEF or HFpEF and its evidence in the manuscript.

Response: A-V fistula-Induced heart failure model exhibits several manifestations of this clinical setting including kidney dysfunction , neurohormonal activation and cardiac hypertrophy.  After 2 and 4 weeks from the induction of the disease rats still in the compensatory phase and considered HFpEF despite the typical characteristics of the disease such as activation of RASS, SNS, ADH, ANP/BNP and impaired kidney function. It  would take 8 weeks and the animals convert to HFrEF .  We referred to this point in the Methods  (Page 21).

Therefore, this model is acceptable one for studying the clinical manifestations of heart failure at its early compensated phase as well as evolved phase depending on the follow up period. 

Reviewer 3 Report

Comments and Suggestions for Authors

I have carefully reviewed the revised manuscript, and while the authors have made some improvements, there are still significant issues that need to be addressed before the manuscript can be considered for publication.

Histological analyses: It is crucial to include histological analyses not only of the kidney but also of the heart. These analyses will provide additional supporting evidence for the data presented in the manuscript and enhance the overall comprehensiveness of the study.

Measurement of AngII: As angiotensin II (AngII) is known to play a critical role in kidney dysfunction, it is essential to measure AngII levels in kidney tissue. This measurement will contribute to the mechanistic understanding proposed in the study and strengthen the hypothesis put forth by the authors.

Author Response

Thank you for your valuable comments. 

We absolutely agree with you about the significance of the renal and cardiac histological analysis of the various experimental groups. Yet, it make take few months to conclude it. We promise to achieve this goal and publish the results in histology oriented paper in IJMS.

Unfortunately, we do not have any more frozen renal tissue of the studied groups as we used the obtained frozen halves kidneys for various molecular analysis.